# Acceptability of smart locker technology for dispensing chronic disease medication among patients and healthcare providers in Nigeria

**Ibrahim Bola Gobir**[1]*, **Piring'ar Mercy Niyang**[2], **Havilah Onyinyechi Nnadozie**[2], **Samson Agboola**[2], **Helen Adamu**[2], **Fatimah Ohunene Sanni**[2], **Angmun Suzzy Otubo**[3], **Idris Saliu**[4], **Adetiloye Oniyire**[5], **Deus Bazira**[1], **Ayodotun Olutola**[1]

1 Center for Global Health Practice and Impact, Georgetown University, Washington, DC, United States of America, 2 Georgetown Global Health Nigeria, Federal Capital Territory, Abuja, Nigeria, 3 Savannah Health System Innovation Limited, Federal Capital Territory, Abuja, Nigeria, 4 Center for Clinical Care and Clinical Research, Federal Capital Territory, Abuja, Nigeria, 5 Johns Hopkins Program for International Education in Gynecology and Obstetrics, Federal Capital Territory, Abuja, Nigeria

* ibg7@georgetown.edu

## Abstract

Smart lockers are automated delivery machines. They have been used in dispensing ARVs and Tuberculosis medication to chronically ill patients in South Africa, Kenya, and Eswatini. However, there is no evidence of smart lockers in dispensing chronic disease medication in Nigeria. This study aimed to assess the acceptability of smart lockers in dispensing chronic disease medication and to describe the barriers to accessing care among patients with chronic diseases medication in 5 states in Nigeria. We conducted a cross-sectional study among healthcare workers and patients living with chronic diseases in five Nigerian states of Adamawa, Akwa Ibom, Cross River, Benue, and Niger between November and December 2021. A total of 1,133 participants were recruited (728 patients and 405 healthcare workers). The results revealed that most patients and healthcare workers agreed that using smart lockers for drug dispensing will lead to reduced transportation costs, hospital waiting times, the workload of healthcare workers, and decongestion of health facilities. The majority of the patients living with chronic diseases (43%) and healthcare workers (51%) showed high acceptability for the use of smart lockers. The use of smart lockers in dispensing chronic disease medication in Nigeria is feasible, and patients and healthcare workers are willing to accept the smart lockers, provided that a patient-centred implementation strategy is developed.

## Introduction

The global burden of chronic diseases is on the rise. Much of this chronic disease burden is from low- and middle-income countries like Nigeria, which already has a high prevalence of infectious diseases. In Nigeria, chronic diseases such as cancer, diabetes, cardiovascular disease, and chronic respiratory disease account for 29% of all deaths. They are projected to become the leading cause of morbidity and mortality by 2030 [1]. This epidemiological

**Data Availability Statement:** Data set supporting the findings of this study are available within the

paper, and its supporting information files. (Supplementary data).

**Funding:** • Initials of Author: Ibrahim Bola Gobir: (IBG) • Full Name of each funder: Georgetown Medical Center Dean of Research • URL of funders website: https://gumc.georgetown.edu/ NO- The funders had no role in study design, data collection, and analysis, decision to publish or preparation of the manuscript.

**Competing interests:** The authors have declared that no competing interests exist.

transition presents the burden of providing longitudinal management of chronic diseases in a weak health system that is more suited for managing acute illnesses. Among others, long waiting times and provider fatigue impact quality are commonplace and adversely affect service delivery. Therefore, strategies to improve chronic disease management include simplifying managerial approaches to drugs and follow-up to make care more client-centred and reduce the burden on the health system. The care differentiation describes these approaches implemented in HIV treatment programs to address the type, frequency, and location of service delivery for clients based on their needs. Some of these best practices in care differentiation are anchored on applying technology. Notable examples include the increasing adoption of telemedicine for medical consultation and follow-up, the utilization of electronic medical records and unique identifiers to enhance access to care from providers at locations of convenience, etc. A relative innovation is using smart lockers, an automated delivery machine to dispense drug refills to patients with chronic illnesses using unique codes within facilities or communities when medical consultation is not required.

Smart lockers promote patient-centred care by enabling quicker and easier dispensing of medications to patients. Its use has reduced waiting times to collect lifesaving medication while reducing foot traffic in overcrowded clinics [2]. These positive outcomes have been recorded in a few Southern African countries where smart lockers were piloted [3]. Nigeria shares similar socioeconomic and health characteristics with these countries and requires similar interventions to simplify care, reduce unnecessary client-provider interactions and reduce provider fatigue. Hence, the need to evaluate the acceptability of smart lockers for drug refills among clients with chronic illnesses and their health providers.

## Methods

### Study design and setting

We used a cross-sectional research design. The data was collected from the administered questionnaires developed for patients and healthcare workers in Nigeria. The participants were recruited from 8th of November to 4th of December 2021 among healthcare workers and persons living with chronic diseases across Adamawa, Akwa Ibom, Cross River, Benue, and Niger states in Nigeria.

We conducted this study in secondary health facilities (S1 Appendix) in 5 states in Nigeria. These facilities provide care for patients with chronic diseases such as Diabetes, hypertension, HIV, cancer, Tuberculosis etc. The facilities were identified via implementing partners supporting clinical care.

A semi structured questionnaire was used. The entry page of the survey contained survey information and objectives. The participants after giving their consent were asked to complete a questionnaire of 37 items. The first section contained the sociodemographic characteristics including gender, age, marital status, and highest educational qualification. Also, the chronic diseases of the participants and the barriers to accessing care among them. The second section assessed the acceptability of smart lockers.

### Study population

The target populations for this study are health workers who provide healthcare services to individuals living with chronic diseases. Persons living with one or more of the following chronic illnesses: HIV, diabetes, hypertension, chronic kidney disease, Cancer, TB, etc.

The inclusion criteria were either giving care or receiving care from the facilities listed in the S1 Appendix.

We obtained written and informed consent from all participants.

## Sample size determination

**Patients of persons living with chronic illnesses.** According to a research paper on patterns of chronic illnesses conducted in Nigeria in 2020 [4], the percentage of chronic diseases in Nigeria is about 64.9% of the population. This was used as a proxy to estimate the population prevalence of chronic diseases given that the population of Nigeria is about 200 million [5]. This was entered into the Raosoft® sample size calculator at 4% margin of error, 95% confidence interval and 50% response distribution to yield a minimum sample size of 601.

**Healthcare workers (HCW).** According to a research paper on health work force estimated between 2016 to 2030 to understand if Nigeria will have enough workforce. [6], the estimated number of HCWs was 621,205. This was used as a proxy for HCWs providing care to chronic illness patients. This was inputted into the Raosoft® sample size calculator at a 5% margin of error, 95% confidence interval and 50% response distribution to yield a minimum sample size of 384.

**Selection of participants.** Healthcare workers providing chronic disease care and patients with chronic illnesses were identified through the selected healthcare facilities in the respective states. Stratified random sampling was used to identify patients living with chronic illness and HCWs that meet the inclusion criteria. The participants were stratified based on gender i.e., 50% male and 50% female, to ensure adequate representation.

## Data sources/Collection methods

**Patients and health workers' survey.** We used a self-administered semi-structured questionnaire to collect sociodemographic characteristics and the acceptability of smart lockers. We also collected information on current barriers to accessing treatment, perception of using the smart lockers for the collection of medication, and potential benefits and challenges with using the smart lockers for collecting medication.

**Statistical analysis.** We performed descriptive statistics to summarize variables and chi-square test was used to determine the associations between variables. All analysis was performed at a 5% significance level and carried out using STATA version 15.0 and Microsoft Excel (2016).

## Ethical considerations

**Informed consent.** The entry page of the survey contained information on the study objectives, eligibility criteria, data privacy and researchers' disclaimers. Informed consent was obtained from participants. If a participant ticks the "I agree" checkbox on the survey, it was considered sufficient to provide informed consent. Entries from participants who do not meet the inclusion criteria was not processed for data analysis.

**Confidentiality.** All entries were recorded anonymously. Personally identifiable information was not collected from the participants. Privacy of the subjects' information was guaranteed.

**Risks and benefits.** There were no adverse effects that affected the right and welfare of the subjects. There were no direct benefits associated with participation in the study.

## Ethical clearance

The study protocol was submitted for review and approval to the Nigeria Health Research Ethics Committee (NHREC). Permission was also obtained from the administrators of the health facilities where data collection was conducted.

## Results

A total of 1,180 persons were invited and 1,133 participants completed the survey giving a response rate of 96% (Fig 1). Of the 1,133 responses included in the analysis, 405 were received from HCWs, and 728 were from persons living with chronic illnesses or caregivers of persons living with chronic diseases in Nigeria.

Table 1 shows the demographic characteristics of the participants. Most of the respondents in the patient survey were female (55.9%), whereas most HCWs were male (51.4%) and 67.9% and 50.1% of the participants were between the age group of 18–35 years and 36–60 years for HCWs and patients respectively; 48.9% and 57.6% of the participants were single for HCWs and married for patients respectively. The majority of the participants had Post-Secondary Education (86.9%) and Secondary Education (40.5%) for HCWs and Patients groups respectively.

The most common morbidity among the participants was HIV (76.7%). Among the 599 participants with HIV, more than half were females (n = 316, 52.8%). Among female participants with HIV, the age group with the highest proportion was 18–35 years (n = 156, 49.4%) while among males the age group with the highest proportion was 36–60 years (n = 135, 55.8%), (Table 2).

Similarly, the healthcare workers were young between ages of 18 and 35 years and the majority were nurse (42.5%), followed by Laboratory scientists (21.5%), Pharmacists (12.6%), and CHEW (10.1%), (Table 3).

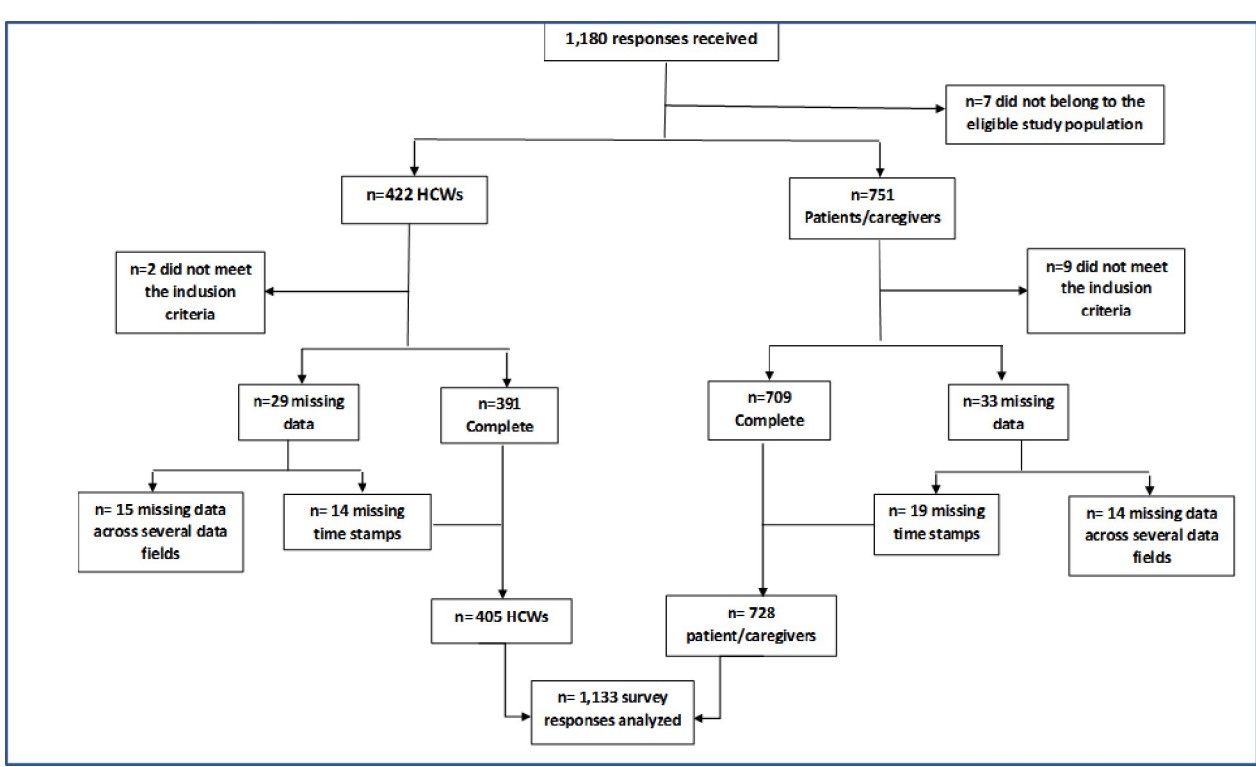

**Fig 1. Flow diagram of the study sample size.**

**Table 1. Demographics of the participants.**

| Variables | HCWs, n = 405 | Patients, n = 728 |
|---|---|---|
| **Gender** | | |
| Male | 208 (51.4%) | 321 (44.1%) |
| Female | 197 (48.6%) | 407 (55.9%) |
| **Age** | | |
| 18–35 | 275 (67.9%) | 338 (46.4%) |
| 36–60 | 125 (30.9%) | 365 (50.1%) |
| > 60 | 5 (1.2%) | 25 (3.4%) |
| **Marital status** | | |
| Single | 198 (48.9%) | 187 (25.7%) |
| Married | 197 (48.6%) | 419 (57.6%) |
| Previously Married | 10 (2.5%) | 122 (16.8%) |
| **Highest educational qualification** | | |
| No formal Education | 2 (0.5%) | 97 (13.3%) |
| Primary Education | 7 (1.7%) | 135 (18.5%) |
| Secondary Education | 44 (10.9%) | 295 (40.5%) |
| Post-Secondary Education | 352 (86.9%) | 201 (27.6%) |

**Table 2. Gender, age, and reported morbidity of patients.**

| | Female | | | Male | | | Total |
|---|---|---|---|---|---|---|---|
| | 18–35 | 36–60 | > 60 | 18–35 | 36–60 | > 60 | |
| **Diabetes** | 7 | 10 | 2 | 7 | 11 | 1 | 38 (5.2%) |
| **Hypertension** | 9 | 20 | 2 | 10 | 11 | 2 | 55 (7.6%) |
| **HIV** | 156 | 148 | 12 | 101 | 135 | 6 | 599 (76.7%) |
| **Cancer** | 4 | 7 | 1 | 4 | 4 | - | 20 (2.8%) |
| **Tuberculosis** | 10 | 13 | 2 | 14 | 8 | 2 | 49 (6.7%) |
| **Others** | 1 | 2 | - | 2 | 2 | - | 7 (1.0%) |

**Table 3. Gender, age, and professional cadre of healthcare workers.**

| | Female | | | Male | | | Total |
|---|---|---|---|---|---|---|---|
| | 18–35 | 36–60 | > 60 | 18–35 | 36–60 | > 60 | |
| Doctor | 8 | 5 | 0 | 7 | 8 | 1 | 29 (7.2%) |
| Nurse | 58 | 17 | 1 | 66 | 30 | 0 | 172 (42.5%) |
| Pharmacist | 16 | 11 | 1 | 16 | 7 | 0 | 51 (12.6%) |
| CHEW | 23 | 12 | 0 | 6 | 0 | 0 | 41 (10.1%) |
| Laboratory scientist | 26 | 7 | 1 | 34 | 18 | 1 | 87 (21.5%) |
| Nutritionist/Dietician | 0 | 0 | 0 | 0 | 1 | 0 | 1 (0.2%) |
| Physiotherapist | 0 | 4 | 0 | 2 | 0 | 0 | 6 (1.5%) |
| Others[2] | 6 | 1 | 0 | 7 | 4 | 0 | 18 (4.4%) |

[2]Care, prevention, and support officer (OVC), Monitoring and Evaluation Officer, ART Linkage and Retention Coordinator, Pharmaceutical Technologist

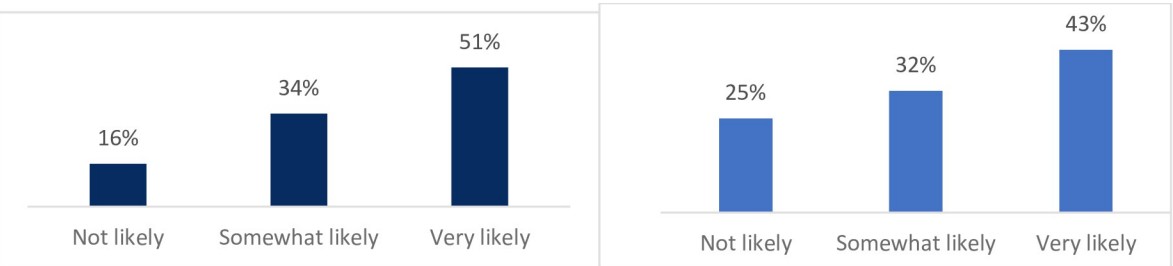

**Fig 2. Acceptability of smart lockers for dispensing chronic disease medication among patients and healthcare workers in 5 states in Nigeria.**

## Acceptability of smart lockers for dispensing chronic disease medication in 5 states in Nigeria

The majority of patients living with chronic disease (51.0%) and healthcare workers (43.0%) were very likely to accept the use of smart lockers for dispensing chronic disease medications (Fig 2).

The proportion of patients with HIV, Hypertension, and Diabetes that indicated they would very likely utilize smart lockers for drug refills was 45.9%, 53.6%, and 74.4%, respectively. The proportions were lower with patients with tuberculosis (36.1%) and cancer (16.0%), (Fig 3).

As shown in Table 4, most people living with HIV and hypertension were likely to accept smart lockers for dispensing medication. The distribution of the acceptance of smart lockers

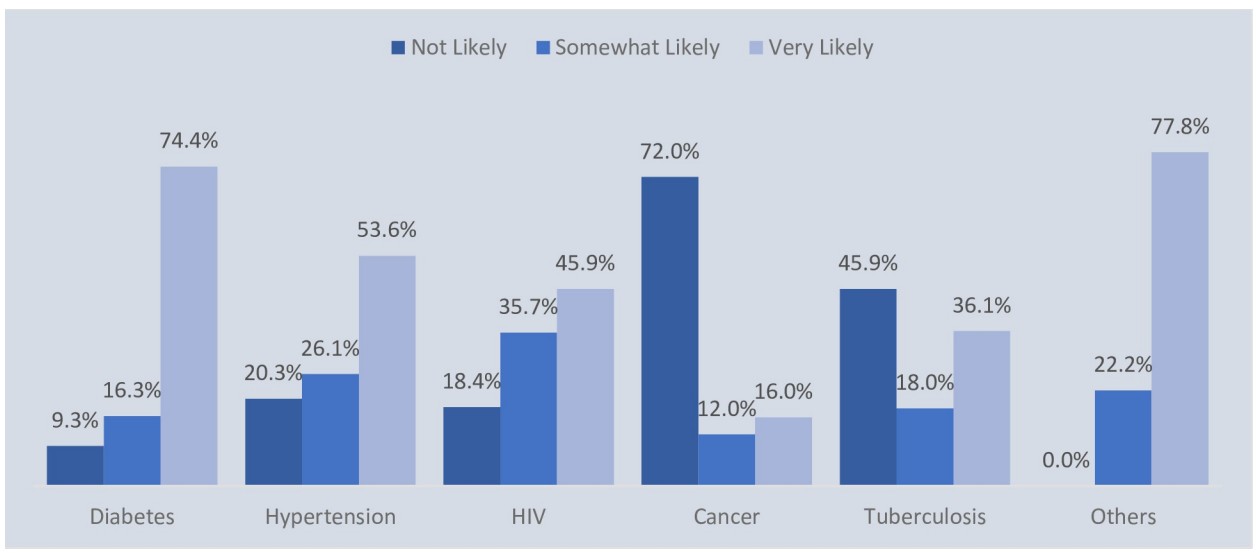

**Fig 3. Acceptability of smart lockers among patients stratified by morbidity in 5 states in Nigeria.**

**Table 4. Acceptability of smart lockers among patients with various chronic diseases.**

|  | Not likely |  | Somewhat likely |  | Very likely |  |
|---|---|---|---|---|---|---|
|  | % | (95% CI) | % | (95% CI) | % | (95% CI) |
| DM | 9.3 | 2.4–29.2 | 16.3 | 4.2–33.7 | 74.4 | 53.7–88.9 |
| Hypertension | 20.3 | 10.2–32.4 | 26.1 | 25.8–40.3 | 53.6 | 39.7–67.0 |
| HIV | 18.4 | 15.3–21.9 | 35.7 | 31.7–39.8 | 45.9 | 41.7–50.1 |
| Cancer | 72 | 45.7–88.1 | 12 | 1.2–31.7 | 16 | 3.2–37.9 |
| Tuberculosis | 45.9 | 32.1–61.9 | 18 | 9.1–33.3 | 36.1 | 22.7–51.5 |
| Others | 0 | 0 | 22 | 6.4–47.6 | 78 | 52.4–93.6 |
| Total | 21 | 18.1–24.2 | 32.1 | 28.8–35.7 | 46.8 | 43.2–50.5 |

[3]Asthmatic, Surgical cases

technology for dispensing chronic disease medication in Nigeria was significantly different ($\Sigma$-calc. = 75.762, $p < 0.05$).

As shown in Table 5, 48.9% of the patients said that they were unwilling to pay to use the smart lockers; however, 23.2% were willing to pay a token to use the lockers, and 27.9% said their willingness to pay would be dependent on the cost charged for the use of the lockers. Using the conversion rates of 410.95 naira per dollar in the year 2021 when the study was conducted, among those willing to pay, a maximum of 200Naira and 200-500Naira were acceptable to 61% and 27.4% of respondents, respectively.

## Barriers to accessing and providing chronic disease care among patients and healthcare workers in Nigeria

To identify perceived barriers to accessing and providing chronic disease care among patients and healthcare workers, participants were asked some key questions pertaining to the current perceived barriers.

The most frequently encountered barrier reported by patients was the cost of transportation to the hospitals, followed by the distance to the healthcare facility where they receive care.

**Table 5. Patients' willingness-to-pay.**

| Would you be willing to pay a token to use the smart lockers for collecting your drugs? |  | % |
|---|---|---|
| No |  | 356 (48.9%) |
| Yes |  | 169 (23.2%) |
| Depending on the cost |  | 203 (27.9%) |
| **How much would you be willing to pay to use the smart lockers for collecting your drugs?** |  |  |
| Naira | Cents/ Dollar | % |
| 100–200 Naira | 0.24–0.48 | 444 (61.0%) |
| 200–500 Naira | 048–1.21 | 200 (27.4%) |
| 500–1000 Naira | 1.21–2.43 | 55 (7.5%) |
| Above 1000 Naira | Above 2 dollars | 22 (3.0%) |
| Didn't specify |  | 8 (1.1%) |

They also indicated that long waiting times were a barrier to accessing care. The patient reported barriers to accessing chronic disease care where transportation cost (41.9%), distance (30.2%), long waiting time (17.9%), medication cost (4.4%), rare drugs (2.9%) and drug stuck out (2.7%). The healthcare workers reported barriers to giving care to patients with chronic diseases were long working hours (23.7%), limited resources in the health facility (20.0%), high volume of patients (18.0%), physical fatigue (17.8%), completing work responsibilities (11.9%) and, emotional fatigue (8.6%).

## Discussion

The study focused on the acceptability of smart locker technology for dispensing chronic disease medication and significant barriers encountered in accessing chronic disease care in Nigeria, which include the cost of transportation and the long distance to access care. These barriers buttress the affordability, accessibility, and availability of primary care in low to middle-income countries like Nigeria [7]. Chronic care management requires expertise that may not be readily available in primary healthcare settings, requiring referral to secondary health facilities that are less accessible due to distance or cost [8]. The cost of health care to the patients, including transportation costs and subscription fees to use the smart lockers, may be an understandable price to pay, as this will address the issue of long waiting hours at the clinic. This would make access to smart lockers affordable to many patients of low socioeconomic status. However, the majority of the patients were reluctant to accept to pay for the use of smart lockers, as reported in our study. On the part of healthcare workers, barriers to providing chronic disease care were predominantly related to the imbalance in workload and existing capacity within the health facilities. The poor working conditions and high workload encountered by healthcare providers lead to a reduction in their motivation [9]. This reduces the quality of patient-provider interaction and the quality of care provided.

The acceptability of smart lockers for dispensing chronic disease medications was high among patients with chronic illnesses and healthcare workers. Though the study had a preponderance of HIV clients due to the study location, the acceptance was comparably high among the patients with HIV infection, Diabetes, and Hypertension. In addition, most respondents indicated that the smart lockers might best serve people living with HIV between the ages of 18–35. The added privacy or confidentiality that the machine offers due to limited human contact may probably contribute to its acceptance for conditions like HIV that are associated with stigma and discrimination. The HIV treatment programs have implemented several models to differentiate care for clients, including facility and community refills (DSD). Therefore, PLHIVs are more accustomed to innovations in drug refills and are required to use their medications for life compared to patients with Tuberculosis or cancer. Likewise, patients with diabetes and hypertension are empowered in the self-management of their diseases for services such as monitoring blood pressure or blood sugar monitoring without visiting healthcare facilities [10]. The use of smart lockers would therefore serve as an adjunct to these methods of drug dispensing, making it easier for patients to access their drugs within the shortest possible time and may account for the high acceptability rates among these patients. The shorter duration of treatment for Tuberculosis and the severity of the symptoms requiring frequent provider evaluation may account for the lower acceptance for Tuberculosis and cancer patients. In addition, there is limited access to resources and knowledge on self-management, especially about the independent management of patients [11].

Respondents revealed the cost of transportation, distance to health facilities and long waiting time as barriers to accessing care, we believe smart lockers have the potential to reduce the turnaround time and limit barriers [12–14]. Most respondents were likely to accept smart

lockers for dispensing medication, especially to improve access to healthcare through a reduction in waiting time.

This expectedly addresses the barriers to access identified in the study. There was less consensus on the ability of smart lockers to improve treatment adherence. This is probably because medication adherence is multifactorial, including the health system, drug-related, or patient factors. Many factors influence treatment adherence [15]. However, only issues related to accessibility and ease of access can be addressed using decentralized models of care such as smart lockers. Healthcare workers agree that smart lockers could decongest healthcare facilities, reduce workload, reduce provider fatigue and enhance the quality of care.

This study is the first to describe the acceptability of smart lockers for dispensing chronic disease medication in Nigeria. The findings are instructive in piloting this new technology for service differentiation.

In addition, the use of surveys may have led to self-selection bias, as the respondents may have an inherently unique characteristic that may have influenced their choices regarding the use of smart lockers. The study has some limitations as most patient responses were received from PLHIV due to the characteristics of the hospitals selected and the fact that the HIV program provides the most extensive organized chronic care programs in those locations, which may limit the generalizability of the findings to other chronic diseases. Furthermore, the cross-sectional nature of this study, using a semi-structured tool may have limited the variety of responses, as participants were limited to the options provided on the questionnaire for most of the questions. Further research may benefit from using qualitative methods to explore the perceptions of under-represented populations of patients living with chronic diseases and healthcare workers to characterize their specific challenges in accessing or providing chronic disease care and acceptability for using smart lockers and estimate the acceptability of smart lockers.

## Conclusion

Smart lockers for dispensing chronic diseases in Nigeria are a feasible option for addressing the barriers encountered by patients with chronic illnesses, particularly PLHIV in Nigeria, such as distance to access care, high cost of transportation, long wait times, increased patient volume, and the high workload for healthcare workers. Its implementation should consider location and acceptability that enhance privacy and confidentiality.

## Supporting information

**S1 Appendix. Distribution of states and facilities.**
(DOCX)

**S2 Appendix. Supplementary data.**
(XLSX)

**S3 Appendix. Study protocol.**
(PDF)

**S4 Appendix. Approval for study protocol.**
(PDF)

## Author Contributions

**Conceptualization:** Ibrahim Bola Gobir, Piring'ar Mercy Niyang.

**Data curation:** Havilah Onyinyechi Nnadozie, Samson Agboola.

**Formal analysis:** Havilah Onyinyechi Nnadozie, Samson Agboola.

**Funding acquisition:** Ibrahim Bola Gobir.

**Investigation:** Piring'ar Mercy Niyang, Angmun Suzzy Otubo, Idris Saliu, Adetiloye Oniyire, Deus Bazira, Ayodotun Olutola.

**Methodology:** Piring'ar Mercy Niyang, Havilah Onyinyechi Nnadozie, Samson Agboola, Helen Adamu, Deus Bazira, Ayodotun Olutola.

**Project administration:** Havilah Onyinyechi Nnadozie.

**Resources:** Piring'ar Mercy Niyang, Angmun Suzzy Otubo, Idris Saliu, Adetiloye Oniyire, Deus Bazira, Ayodotun Olutola.

**Supervision:** Piring'ar Mercy Niyang.

**Validation:** Piring'ar Mercy Niyang.

**Writing – original draft:** Ibrahim Bola Gobir, Havilah Onyinyechi Nnadozie.

**Writing – review & editing:** Piring'ar Mercy Niyang, Helen Adamu, Fatimah Ohunene Sanni, Deus Bazira, Ayodotun Olutola.

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
