## [Decision Letter · Decision Letter 0]

8 Sep 2023

PONE-D-23-21767Feasibility and Acceptability of Smart lockers Technology for Dispensing Chronic Disease Medication in NigeriaPLOS ONE

Dear Dr. Bola Gobir,

Thank you for submitting your manuscript to PLOS ONE. After careful consideration, we feel that it has merit but does not fully meet PLOS ONE’s publication criteria as it currently stands. Therefore, we invite you to submit a revised version of the manuscript that addresses the points raised during the review process.

ACADEMIC EDITOR: The study findings are of immense value in strengthening patient-centered care especially in this era of digital health. However, the paper requires substantial improvement to ensure clarity of the great message the study is conveying. Please address the following comments in addition to the comments provided by the two reviewers. Also ensure that tables, figures and format are inline with plosone criteria:Financial disclosure:Please be specific about the funding source. If the study wasn’t funded, please state so. If funded, but the funders have no role to play, then indicate the funding source and clarify as such.Selection of facilities – any criteria?Please include the following references Faronbi, Ademuyiwa and Olaogun (2020)Adebayo, Labiran, Emerenini and Omoruyi (2016),
Please specifically define the inclusion/exclusion criteria for the 2 – 3 study groups (patients, relatives, and healthcare providers).Also try to be consistent – patients/care givers or just patients to avoid confusion.Was the study conducted in HIV clinics? Majority of the patients are HIV clients. This indicates the likelihood of targeting HIV clinics. Also do other patients with other chronic illnesses have comorbidity with HIV?Also only patients are represented in the results but in the study population it was stated that patients/caregivers. Will be good to disagregate between these 2 groups.Figures 2 and 3 titles should be specific. If the decision is to generalize this study (Nigeria wide), then the titles should include Nigeria.The statistical analysis may be misleading because there are no CI especially for the chronic diseases since there is huge margin between the disease groups (absolute numbers). Closely related to this concern is the generalization of the findings (Nigeria wide) which may be misleading since the 5 states are not representative of the country based on socio-economic disparities.Willingness to pay – based on this, it will be highly important to define “acceptability” in the context of this study. Is “willingness to pay” = “acceptability” in the study context or is among the acceptability criteria? Also, can Naira equivalent be included at the time of the study in USD?“Barriers to accessing and providing chronic disease care among patients and healthcare workers in Nigeria” – this subheading is “hanging”. Will be good to introduce this in the methodology.“Feasibility” is not defined. Will be good to add a section about this and to clarify how it was assessed. Otherwise, it shouldn’t be part of the “title” and shouldn’t be mentioned in the paper.Brief description of the questionnaire and relevant variables in the methods will be helpful in addressing some of the comments above==============================

We look forward to receiving your revised manuscript.

Kind regards,

Ibrahim Jahun, MD, MSC, PhD

Academic Editor

PLOS ONE

4. Please include a copy of Tables 6 and 8 which you refer to in your text on pages 10 and 11.

Reviewers' comments:

Reviewer's Responses to Questions

**Comments to the Author**

1. Is the manuscript technically sound, and do the data support the conclusions?

Reviewer #1: Yes

Reviewer #2: Yes

2. Has the statistical analysis been performed appropriately and rigorously? 

Reviewer #1: No

Reviewer #2: Yes

3. Have the authors made all data underlying the findings in their manuscript fully available?

Reviewer #1: Yes

Reviewer #2: Yes

4. Is the manuscript presented in an intelligible fashion and written in standard English?

Reviewer #1: Yes

Reviewer #2: Yes

5. Review Comments to the Author

Reviewer #1: The paper has satisfactorily addressed the problem statement. The manuscript was well written, and data was made available during the review. Mixed citation style was observed as well as missing referenced. The sample size for the participant might be overestimated considering the geographical coverage of the study. Additional comments wer uploaded.

Reviewer #2: The authors research aims to introduce a more secured electronic storage system with capability for automated delivery functionalities to dispense chronic disease medication or drug refills to patients with chronic illnesses assessing healthcare services within facilities or communities, using unique codes. As a result, overcrowding, patients’ out of pocket payment to and from, and waiting time in clinics are reduced. This technology also serves to support patient-centered healthcare service delivery model and reduces healthcare service providers’ fatigue (pp. 3-4). This research revealed that majority of the HCWs, the patients and their caregivers accepted the use of this technology (p.2) if patient centered. However, it would have been better if the authors maintain consistency in stating the actual number of Nigerian states that participated in this research. In some places, 6 states were mentioned, another place stated 5 states (See abstract page, Method section p. 4, line 67). I wished the authors rephrase the sentences in lines 198 – 201 0f page 12) for clarity. This research has a potential to improve a differentiated model of patient care, especially amongst chronically ill patients if well implemented. On the other hand, it will work best amongst literate and technology savvy populations but may pose some form of challenges amongst the unlettered or the population without good knowledge on high-end technology-oriented devices.

6. PLOS authors have the option to publish the peer review history of their article (what does this mean?). If published, this will include your full peer review and any attached files.

Reviewer #1: **Yes: **Mukhtar Liman AHMAD

Reviewer #2: No

---

## [Author Response · Author response to Decision Letter 0]

21 Oct 2023

Reviewer 1(Mukhtar Liman Ahmed): Thank you for your comment on number of states. It has been clarified and addressed: 5 states were included in the study according to Appendix 1 and the result.

Reviewer 1(Mukhtar Liman Ahmed): It is deeply appreciated that you pointed out the missed references; we have now included these references in the reference list using the PLOS ONE reference style.

Reviewer 1(Mukhtar Liman Ahmed): Thank you for pointing the mixed references, they have been cited properly and the reference style adjusted accordingly and consistently.

Reviewer 1(Mukhtar Liman Ahmed): Thank you for this feedback, the sentence has been revised and rephrased to be adjusted in line with the result of the study to improve clarity.

Reviewer 1(Mukhtar Liman Ahmed): Thank you for pointing this out based on the mixed number of states and the sample sizes based on the national estimated parameters. The study was conducted in 5 states, and we have corrected this. We agree that using national estimates may not be ideal. However, we used these estimates because we believe national estimates may be appropriate for a multi-state study than using single study estimates. We oversampled to account for the inherent nature of poor response rates that have been traditionally reported for online surveys.

Reviewer 1(Mukhtar Liman Ahmed): Thank you for your keen observation on the rational for the error margins. We varied the error margins to adjust for differences in expected precision based on the calculated sample sizes of the two study populations.

Reviewer 2(Ibrahim Jahun, Academic editor): Thank you for your comment on financial disclosure. The study was funded by Georgetown University Medical Center, Dean of Research. The funder did not play any role in the research. This has been addressed in the cover letter and the section allotted for funding information in the submission form.

Reviewer 2(Ibrahim Jahun, Academic editor): Thank you for your comment on the criteria for facility selection. The facilities were identified and randomly selected via implementing partners supporting clinical care.

Reviewer 2(Ibrahim Jahun, Academic editor): Thank you for bringing our attention to the missing references. They have been included.

Reviewer 2(Ibrahim Jahun, Academic editor): This comment is well received on the inclusion criteria for our study group. The inclusion criteria for the study were individuals who are either or receiving care from facilities listed in Appendix 1”. We collected information from healthcare providers and patients. For a few of the patients who were not able to complete the survey by themselves, this was completed by their relative on their behalf. The relatives were not part of the study population and we have updated that.

Reviewer 2(Ibrahim Jahun, Academic editor): This comment on patients/ care providers is appreciated and noted. Our document reflects “patients” going forward.

Reviewer 2(Ibrahim Jahun, Academic editor): Thank you for your question on whether the study was conducted in HIV clinics. The study was conducted among patients with chronic illnesses and Health Care Workers which were identified through ART programs’ implementing partners (IP). However, for this study, we included patients from other general and disease-specific clinics within the same hospitals. We did not specifically set out to screen for HIV in other clinics. However, we agree that there is a possibility of comorbidity with HIV in other non-HIV clinics that patients may be unaware of.

Reviewer 2(Ibrahim Jahun, Academic editor): This is kindly noted and adjusted. Our document reflects “patients” going forward. 

Reviewer 2(Ibrahim Jahun, Academic editor): This comment is kindly noted. This adjustment has been made in this section of the result to include “in 5 states in Nigeria.” 

Reviewer 2(Ibrahim Jahun, Academic editor): We have included the proportions and 95% CIs of the proportions for acceptability across different chronic diseases. 

Reviewer 2(Ibrahim Jahun, Academic editor): Thank you for your observation on the lack of naira equivalent to USD. Yes, willingness to pay gives further justification for the acceptability of smart lockers. The Naira equivalent to USD at the time of the study has been included.

Reviewer 2(Ibrahim Jahun, Academic editor): thank you for sharing your observation on the hanging subheading “Barriers to accessing and providing chronic disease care among patients and healthcare workers in Nigeria”. This is kindly noted. This section has been addressed in the objectives, results and the discussion section.

Reviewer 2(Ibrahim Jahun, Academic editor): We agree with the reviewer that we did not define feasibility in this paper. We did not set out to assess feasibility on its own but indirectly. Our aim was to directly measure acceptability of and willingness-to-pay for smart locker and indirectly use them as indicators of feasibility of this intervention in Nigeria.

Reviewer 2(Ibrahim Jahun, Academic editor): The comment on defining feasibility is well received. We have now updated the title to reflect this: “Acceptability of smart locker technology for chronic disease medication among patients and healthcare providers in Nigeria.”

Reviewer 2(Ibrahim Jahun, Academic editor): Thank you for your comment on description of questionnaire. This is noted and has been addressed and a description of the questionnaire has been included in the methods section.

Additional Clarifications 

1. The PLOS ONE File naming format was used as requested. 

2. The PLOS ONE questionnaire on Inclusivity in global research is not required for this paper as this applies to researchers who travelled to a different country to conduct research.

3. The full ethics statement of our study has been included in the methods section, as requested.

4. The tables referred to (6 & 8) on pages (10 &11) have been merged into tables (4 & 5) based on relevant information required for the tables. 

5. All in-text citations have been updated to match accordingly.

---

## [Decision Letter · Decision Letter 1]

13 Nov 2023

Acceptability of smart locker technology for dispensing chronic disease medication among patients and healthcare providers in Nigeria.

PONE-D-23-21767R1

Dear Dr. Gobir,

We’re pleased to inform you that your manuscript has been judged scientifically suitable for publication and will be formally accepted for publication once it meets all outstanding technical requirements.

Kind regards,

Ibrahim Jahun, MD, MSC, PhD

Academic Editor

PLOS ONE

Additional Editor Comments (optional):

Reviewers' comments:

Reviewer's Responses to Questions

**Comments to the Author**

1. If the authors have adequately addressed your comments raised in a previous round of review and you feel that this manuscript is now acceptable for publication, you may indicate that here to bypass the “Comments to the Author” section, enter your conflict of interest statement in the “Confidential to Editor” section, and submit your "Accept" recommendation.

Reviewer #1: (No Response)

Reviewer #2: All comments have been addressed

2. Is the manuscript technically sound, and do the data support the conclusions?

Reviewer #1: (No Response)

Reviewer #2: Yes

3. Has the statistical analysis been performed appropriately and rigorously? 

Reviewer #1: (No Response)

Reviewer #2: Yes

4. Have the authors made all data underlying the findings in their manuscript fully available?

Reviewer #1: (No Response)

Reviewer #2: Yes

5. Is the manuscript presented in an intelligible fashion and written in standard English?

Reviewer #1: (No Response)

Reviewer #2: Yes

6. Review Comments to the Author

Reviewer #1: (No Response)

Reviewer #2: The authors have done justice to the previous review comments and have provided all relevant supporting Documents. Regarding the changes made to the manuscript initial title page, I will suggest the Authors revert to the National Health Research Ethics Committee of Nigeria (NHREC) to inform of the changes in title for easy referencing in future.

7. PLOS authors have the option to publish the peer review history of their article (what does this mean?). If published, this will include your full peer review and any attached files.

Reviewer #1: **Yes: **Mukhtar Liman Ahmed

Reviewer #2: **Yes: **Bassey, Orji Orji

---

## [Editor Report · Acceptance letter]

17 Nov 2023

PONE-D-23-21767R1 

Acceptability of smart locker technology for dispensing chronic disease medication among patients and healthcare providers in Nigeria. 

Dear Dr. Gobir:

I'm pleased to inform you that your manuscript has been deemed suitable for publication in PLOS ONE. Congratulations! Your manuscript is now with our production department. 

Kind regards, 

on behalf of

Dr. Ibrahim Jahun 

Academic Editor

PLOS ONE